# Effect of neutral winds on the creation of non-specular meteor trail echoes

Freddy Galindo[1], Julio Urbina[1], and Lars Dyrud[2]

[1]Communications and Space Sciences Laboratory, Department of Electrical Engineering, The Pennsylvania State University, University Park, PA, USA.
[2]OmniEarth, Inc. 2015; 251 18th Street South - Suite 650, Arlington, VA, USA.

**Correspondence:** J. Urbina (jvu1@psu.edu)

**Abstract.** Non-specular meteor trail echoes are radar reflections from plasma instabilities that are caused by field-aligned irregularities. Meteor simulations are examined to show that these plasma instabilities, and thus the associated meteor trail echo, strongly depend on the meteoroid properties and the characteristics of the atmosphere in which the meteoroid is embedded. The effects of neutral winds, as a function of altitude, are analyzed to understand how their amplitude variability impacts the temporal-space signatures of non-specular meteor trail echoes present in VHF radar observations. It is found that amplitudes of the total horizontal neutral wind smaller than 0.6 m/s do not provide the right physical conditions to enable the genesis of non-specular meteor echoes. It is also found that a 0.0316 $\mu$g meteoroid traveling at 35 km/s can be seen as a meteor trail echo if the amplitudes of horizontal neutral winds are stronger than 15 m/s. In contrast, a 0.316 $\mu$g meteoroid, traveling at the same speed, requires horizontal winds stronger than 1 m/s to be visible as a meteor trail echo. The neutral velocity threshold illustrates how simulations show that no trail echo is created below a critical wind value. This critical wind value is not mapped directly to radar observations but it is used to shed light on the physics of meteor trails and improve their modeling. The meteor simulations also indicate that time delays on the order of hundreds of milliseconds or longer, between head-echoes and non-specular echoes, which are present in VHF backscatter radar maps can be a consequence of very dense plasma trails being affected by weak horizontal neutral winds that are smaller than 1 m/s.

## 1 Introduction

Day and night, meteoroids smaller than a grain of sand penetrate the Earth's upper atmosphere and generate meteor plasma trails at altitudes between 70 and 140 km. Meteor research is typically conducted with photographic and TV cameras and specular meteor radars. Specular radars detect echoes from the trail of ionization formed perpendicular to the radar **k** vector by a meteoroid during atmospheric entry. This specular condition requires that only trails formed perpendicular to the radar **k** vector reflect strongly without destructive interference (Ceplecha et al., 1998).

For more than two decades, scientists have studied two new types of radar meteor reflections. These reflections are known as meteor head-echoes and non-specular trails, typically observed with high power and large aperture (HPLA) radars. Non-specular meteor echoes are radar reflections from meteor plasma instabilities that are generated from field-aligned irregularities (FAI), i.e., non-specular meteor echoes are detected when the radar is pointed perpendicular to the Earth's magnetic field **B**

(Chapin and Kudeki, 1994; Zhou et al., 2001; Oppenheim et al., 2001; Dyrud et al., 2002; Oppenheim et al., 2003, 2014; Chau et al., 2014; Oppenheim and Dimant, 2015; Dimant and Oppenheim, 2017a, b; Sugar et al., 2018, 2019). Although substantial progress has been made in meteor trail studies, we still do not understand the degree to which meteoroid and atmospheric properties affect the creation of non-specular meteor trail echoes. Specifically, this paper investigates the effect of horizontal atmospheric winds on the development and progression of non-specular meteor trails. Our goal is to understand how a realistic

vertical wind profile will influence echo structures routinely seen in VHF range-time intensity maps. Assessing the effects of neutral wind profiles on non-specular meteor trails are essential for scientific studies of meteors, the Earth's Upper Atmosphere, and engineering applications of meteors (Dyrud et al., 2005; Dyrud et al., 2007; Malhotra et al., 2007). To determine the effect of a neutral wind profile on non-specular trail progression; we augmented the physical model of the evolution of an individual meteor from direct atmospheric entry to trail instability and diffusion (Dyrud et al., 2005; Dyrud et al., 2007) by including the

most recent empirical climatological models with the corresponding ionospheric and atmospheric parameters.

The work reported here, and the model we have constructed seeks to understand how a meteoroid of a given mass, velocity, and entry angle will produce different non-specular radar signatures due to different values of horizontal neutral winds. To characterize these effects, we present artificial radar range-time-intensity (RTI) images over Salinas, Puerto Rico, with different meteor properties such as mass and velocity. These simulations are further compared with actual radar observations of

non-specular meteor trail echoes collected over Salinas, Puerto Rico; during the Coqui-II NASA Rocket Campaign (Urbina et al., 2000). This work complements the research described in Hinrichs et al. (2009) and Dyrud et al. (2011) and effectively demonstrates that non-specular meteor trail echoes can exhibit unique radar signatures due to the variability of horizontal neutral winds. The implications of our simulations are extended to interpret meteor trail data collected with the 50 MHz Jicamarca radar (Chau and Galindo, 2008). Unlike the radar used during Coqui-II Campaign, Jicamarca radar usually, but not always,

observes both meteor head-echoes and their associated non-specular meteor trail echoes. These two types of meteor echoes provide a unique opportunity to correlate changes in the meteor head-echo with variations in the non-specular echo, and therefore help to determine whether the radar signature in the trail echo is generated by the parent meteoroid or the background atmosphere.

This paper is organized as follows. In Section 2, we describe the new features of the Non-Specular Meteor Trail Echo

Simulator. In Section 3, we analyze the effect of horizontal neutral wind on the progression of meteor trails, in particular, the importance of horizontal neutral winds sustaining plasma instabilities in the meteor trail. Section 4 presents and discusses radar observations of meteor trail echoes that resemble meteor simulations that capture the effect of neutral winds while Section 5 summarizes the conclusions of our studies.

## 2   Non-Specular Meteor Trail Echo Simulator

The non-specular meteor trail echo simulator (NSMES) is fully described in Dyrud et al. (2005) and Dyrud et al. (2007) but has been updated to include the most recent climatological models for the atmospheric and ionospheric drivers and background conditions that we report in this paper. As explained in these papers, the model starts by computing the amount of ablated

particles created behind the meteoroid body. These energetic particles are then used to calculate amount of ionization created in the trail. Here we assume that the ionization created in the trail is initially distributed in a cylindrical volume defined by the initial radius. At this point, trail is expanded by either ambipolar diffusion or turbulent diffusion to simulate the absence or presence of plasma instabilities in the trail during its evolution (Dyrud et al., 2001; Yee and Close, 2013). The plasma instability analysis is based on meteor Farley-Buneman Gradient-Drift (FBGD) instability reported in Dyrud et al. (2002), Oppenheim et al. (2001), and Oppenheim et al. (2003). NSMES assumes that a non-specular meteor trail echo is created because the trail becomes Bragg reflective at altitudes where plasma instabilities can develop (Dyrud et al., 2002). The results of the simulations produce artificial radar range-time-intensity (RTI) images that we use to compare with Coqui-II and Jicamarca meteor observations.

The atmospheric/ionospheric parameters needed to execute NSMES include electron density, atmospheric neutral mass density, neutral temperature, and neutral winds; for a specific location and time. These numerical values of the properties of the atmosphere are obtained from the latest versions of these climatological models: NRLMSISE-00 empirical model of the atmosphere (Picone et al., 2002), Horizontal Wind Model HWM14 (Drob et al., 2008, 2015), and the International Reference Ionosphere IRI-2016 (Bilitza and Reinisch, 2008; Bilitza, 2018). Our model and the associated software can be executed in a general-purpose PC-based system. It can easily be adapted and combined with other tools to study very large meteor populations. In contrast, as far as we know, more sophisticated 3D meteor models require supercomputer clusters and do not fully simulate the actual extent of a meteor trail or produce results that can be closely compared to 2D observational data. Although our numerical model is a simplified representation of the meteor physics, it can produce very good and fine details such as those reported in this paper. Our model can be used to account for and understand the statistical outcome of thousands of meteors acting collectively on the Earth's Upper Atmosphere.

Figure 1 displays an example of an artificial RTI, produced using NSMES; showing a simulated head-echo and the non-specular meteor trail. In this plot, meteor line density in units of electron per micron for the thin head-echo is shown in the vertical color bar. The non-specular trail color indicates FBGD instability growth rate in units of inverse seconds. The meteor generating this head-echo and non-specular trail had a known mass of $1\mu g$, traveling at 30 km/s, at a zenith angle of $45°$. Vertical profiles of the corresponding atmosphere/ionosphere parameters covering the same altitude range for the simulations are presented in Figure 2. These climatological properties were chosen to characterize the atmosphere and ionosphere over Salinas, Puerto Rico, at 00:00 UT on January 1, 1998. Of these three vertical profiles, the horizontal neutral wind vertical-profile; is the only atmospheric parameter used by NSMES that exhibits more amplitude variability in altitude. This characteristic of neutral winds influences the structure of the resulting meteor trail echo, if for instance, neutral winds value drastically change in a region that is smaller or comparable in size of the meteor trail. These results are further discussed and presented in the following section.

## 3 Effect of Horizontal Neutral Winds on the Space-Time Evolution of Non-Specular Trails

We deployed a similar simulation approach described in Hinrichs et al. (2009) and Dyrud et al. (2011) but focus specifically on the impact of a more realistic vertical profile of horizontal neutral wind on the space-time meteor trail evolution at a specific mid-latitude Location: Salinas, Puerto Rico. We kept the main elements and considerations of their simulation procedures but focused on two specific case studies to understand how the amplitude variability of neutral wind values affects the duration of non-specular meteor trail echoes.

Since we are interested in comparing trail durations between simulations and observations, we computed simulated trail duration at a given altitude using this simple criteria; subtracting time values between the longest duration of non-specular echo and the corresponding onset of head-echo/non-specular echo on an artificial RTI image, as illustrated in Figure 1. For example, for this figure, the trail duration is about 7.5 s, at an altitude of 97.5 km. The corresponding range span of the non-specular echo is approximately 10 km.

To assess the impact of horizontal neutral winds on the time duration of non-specular meteor trail echoes, simulations were produced using horizontal neutral winds between 2.5 m/s and 100 m/s, meteoroid masses between $3.16 \times 10^{-9}$g and $3.16 \times 10^{-5}$g, and speeds between 11 km/s and 72 km/s. We used these meteoroid parameters because they represent a commonly detected class of meteoroids (Mathews et al., 2001; Dyrud and Janches, 2008). Additionally, NSMES assumes that meteoroids have a chondritic composition (45% oxygen, 15% iron, 9% magnesium, and 31% silicon) with a mean atomic

mass weighing about 30 amu and have an entry zenith angle of $45°$. Our analysis also employed the atmospheric properties shown in Figures 2a and 2c. These two figures display typical vertical amplitudes of neutral mass and electron densities used in the simulations reported on this paper, which were used for a case study of meteoroids traveling with a known velocity of 35 km/s. Figure 2b represents the vertical amplitude variability of neutral winds. This specific vertical profile was used to produce meteor simulations shown in Figure 1.

The outcomes of the case study are displayed in Figures 3a and 3b. Several general conclusions can be inferred upon further examination of these two figures. For example, from Figure 3a, it is clear that meteoroids with small masses require stronger neutral winds to create non-specular echoes. This means that meteor trails with low electron densities need stronger winds to keep the total electron drifts large enough to produce and sustain plasma instabilities and therefore enable the genesis of non-specular meteor trail echoes. Additional details about the meteor physics can be found in (Oppenheim et al., 2001; Dyrud

et al., 2002; Oppenheim et al., 2003). Further inspection of Figure 3a shows that a 0.01 $\mu$g meteoroid can create non-specular echoes if winds are stronger than 60 m/s, while a 0.0316 $\mu$g meteoroid requires winds stronger than 15 m/s to give rise to non-specular echoes. It can also be seen from Figure 3a that if these meteoroids that are traveling at 35 km/s, and have masses smaller than 0.01 $\mu$g, they will never produce non-specular echoes. We repeated extensive simulations of meteoroids traveling with other velocity values. Still, we kept the same range of mass values and neutral winds to determine if other trends could

emerge by changing velocity values. But the results of these simulations produced similar patterns to those depicted in Figure 3a. For instance, if a meteoroid travels at 15 km/s, masses larger than $1\mu$ g are required to create non-specular meteor trail

echoes for the same range (between 2.5 m/s and 100 m/s ) of horizontal neutral winds. In comparison, a meteoroid that travels at 55 km/s needs masses greater that $0.1\mu$g to produce non-specular meteor trail echoes with similar neutral wind ranges.

Another important implication of our meteor simulations is that trail duration linearly increases as a function of the amplitude of horizontal neutral winds to power 0.4 for the corresponding range of meteoroid masses mentioned earlier. This relationship became apparent when plotting trail duration as a function of neutral winds and produced a square root signature. The exponent 0.4 was determined by choosing a value that minimized the root-mean-square-error of the curves. Three examples of simulated non-specular trail duration as a function of horizontal neutral winds to the power 0.4 that corresponds to meteoroid masses of 3.16 $\mu$g, 1 $\mu$g, and 0.31 $\mu$g are shown in Figure 3b. This empirical relationship between trail duration and neutral wind amplitudes provide insight to formulate practical and straightforward equations that could easily give an approach to determine meteor trail duration when considering different mass scenarios. This outcome is complementary to other relationship between non-specular echoes, and the meteoroid mass and speed that are provided in Dyrud et al. (2011).

Perhaps the most important result of these simulations can be expressed with this probing question: is there a minimum value of horizontal neutral wind to enable non-specular meteor trail echoes? We think the answer is "yes," at least for the range of values of meteoroid masses and speeds presented in this paper. We conducted extensive simulations with similar meteoroid parameters and climatological atmospheric conditions as those that produced the results exhibited in Figure 3a but restricted neutral wind values between 0.2 m/s and 10 m/s. Some of the outcomes of these simulations are shown in Figure 4. In this example, the meteor trail is traveling at 35 km/s with masses between $3.16 \times 10^{-9}$g and $3.16 \times 10^{-5}$g. It can be seen that horizontal neutral wind values smaller than 0.6 m/s do not favor the development of non-specular echoes. Some of the implications of these results are further discussed in the next section.

## 4   Discussion

The first important implication of our analysis is captured in Figure 5a. This artificial RTI shows a simulated head-echo and non-specular meteor trail with the same physical characteristics of the parent meteoroid as those that were used to produce Figure 1. The corresponding vertical profiles of the atmosphere/ionosphere parameters covering the same altitude range of these simulations are similar to those in Figures 2a and 2c, but the corresponding vertical profile of the horizontal neutral wind is displayed in Figure 5b. These climatological properties were chosen to characterize the atmosphere and ionosphere over Salinas, Puerto Rico, at 00:00 UT on February 2, 1998. Basically, we determined that multiple non-specular echoes can be created from the same parent meteoroid mass as it makes its journey through a vertical-horizontal wind profile that contains very sharp wind gradients as those depicted in Figure 5b. The absence of non-specular meteor trail echoes near 96 km altitude in the RTI of Figure 5a is compatible with very low wind values that resemble wind shear effects near 96 km, as shown in Figure 5b. Notice that neutral winds values are very small near 96 km but their magnitude rapidly increases below or above this altitude to sustain plasma instabilities. Our result suggests that careful consideration of neutral wind effects on non-specular meteor trail evolution must be considered when interpreting meteor radar reflections. Ignoring neutral wind effects could lead

to misinterpretation of these echoes signatures and associated them, for example, with antenna beam side-lobes or physical fragmentation of meteoroids.

The second implication of our studies is related to the time delay between head-echo and non-specular meteor trail. One example showing this effect is presented in Figure 6. This RTI plot shows a head-echo and non-specular meteor trail echo that were simulated using a 5 $\mu$g meteoroid traveling at 20 km/s. The corresponding atmospheric properties used in this simulation are shown in Figures 2a and 2c. Still, the amplitude of the vertical profile of the neutral wind is maintained constant and equal to 2.2 m/s. We chose this low value of neutral wind to demonstrate that it is possible for non-specular meteor trail echo to take at least 500 ms to be created after the meteoroid has traveled through such low, neutral wind values. This effect is clearly illustrated in Figure 6. It indicates that studies of non-specular meteor echoes as a function of atmospheric properties are critical when interpreting observational meteor reflections obtained with radar sensors.

We just described two scenarios that can manifest when neutral winds amplitudes are close to a critical value that is needed to sustain meteor plasma instabilities, which in turn will create non-specular meteor reflections. These results indicate that the impact of neutral winds on trail evolution is a plausible and complementary explanation to the characteristics exhibited by VHF radar reflections as those reported in (Close et al., 2004; Malhotra et al., 2007; Dyrud et al., 2007; Malhotra and Mathews, 2009; Sugar et al., 2010).

Further, our simulations here are compared with non-specular meteor echoes collected during the Coqui-II campaign (Urbina et al., 2000). We extend the outcomes of our simulations to analyze head-echoes and non-specular meteor echoes detected with the Jicamarca VHF radar during the $\eta$-Aquarids campaign (Chau and Galindo, 2008).

## 4.1  Effects of Neutral Wind Shears on the Characteristics of Non-specular Echoes

As discussed above, the meteor simulations presented in Figure 5 indicate that plasma instabilities can not be sustained in the trail when horizontal neutral winds are weaker than a critical value. The nature of this behavior is because electron drifts plasma instabilities at these altitudes are mainly composed of diamagnetic drifts that are divergence-free and therefore do not contribute to driving plasma instabilities (Oppenheim et al., 2001; Dyrud et al., 2002; Oppenheim et al., 2003). An observational VHF backscatter power map showing a head-echo and non-specular meteor echoes over Salinas, Puerto Rico at 06:15 LT on February 22, 1998, is depicted in Figure 7a. The main features of this experimental RTI are in good agreement with the simulation results presented in Figure 5. A geometrical analysis between the backscatter power and the antenna beam-width of the VHF radar demonstrates that the absence of backscattering echoes in the RTI is not due to the effect of the nulls of the antenna pattern. We also discarded fragmentation effects since there are not interference signatures visible at other ranges. Therefore, we think that wind shears are the most probable cause of the backscatter structures displayed in Figure 7a.

Another VHF backscatter power map of head-echo and non-specular echoes is shown in Figure 7b. This RTI was collected at 01:22 LT on February 23, 1998, over Salinas, Puerto Rico. We conducted more than 1000 meteoroid simulations under different horizontal neutral wind profiles, meteoroid mass and speed ranges that were mentioned in Section 3, and concluded that the meteor reflections shown in Figure 7b could be reproduced using neutral wind shear conditions with a valley that expands over a couple of kilometers in range, near 150 km, and a meteoroid mass of about 0.32$\mu$g, traveling at 30 km/s.

Similarly to Figure 7a, an analysis of the backscatter power of both head-echo and the trail from Figure 7b revealed that the abrupt cessation of non-specular trails near the 150 km range cannot be reproduced with sharp change values in trail densities or antenna beam-width nulls. Thus, the winds condition described lines above represent a viable explanation.

We took a step further and expanded our meteor simulation results to analyze meteor trails collected with the High Power Large Aperture Jicamarca VHF radar. Since this radar has more power than the radar used during the Coqui-II Campaign, it can routinely detect many pairs of head-echo and non-specular trail echoes. For example, on May 6th, 2007, between 04:45 LT and 05:10 LT, the Jicamarca radar collected 17 (out of 103) pairs of head-echo and non-specular trails that displayed an absence of backscatter echoes near 104.5 km altitude. Two of these types of events are shown in Figures 8a and 8b. Routine meteor interferometry analysis, described in (Chau and Galindo, 2008), was applied to the events presented in Figures 8a and 8b. We analyzed each of the head echo events using both received SNR and interferometry analysis. We discarded noise level as a potential explanation since in the examples we report, all trail echoes were at least 3 dB above the noise level. So statistically, it is improbable that noise is responsible for gaps in different echoes at the same range and around the same time. We also discarded antenna nulls as a possible explanation for the trails' gaps since interferometry analysis placed these events in the main lobe of the antenna. These examples from Jicamarca have echo gaps observed only in the trail echoes, as shown in Figure 8a. Notice that there is no drop in power intensity for the head echo around the 104.5 km range when zoom-in in this figure. The head-echo event shown in Figure 8a started at 04:46 LT and was moving with a speed of 65 km/s in a direction almost parallel to the geographic East-West direction. The event shown in Figure 8b started at 04:54 LT and was moving with a speed of 56 km/s in the North-West direction. A full analysis of the 17 observational meteor events revealed meteoroids with different properties of mass, speed, or direction. Since neutral and electron densities typically do no exhibit abrupt changes in altitude, it is reasonable to conclude that neutral wind shears with very low amplitudes are the most probable cause for the absence of backscatter reflections near 104.5km as illustrated in these two events of Figures 8a and 8b. Since the events shown in Figure 8 last less than 3 seconds and therefore the method described in Oppenheim et al. (2009) to estimate neutral winds would not work, we expect to carry out both uncoded and coded radar experiments using Jicamarca High-Power Large Aperture radar and an all-sky meteor radar, to compute neutral wind amplitudes using meteor trails similar to the approach described in (Oppenheim et al., 2009, 2014) and in Li et al. (2012) to establish a complete understanding and characterization of non-specular echoes.

Malhotra and Mathews (2009) reported meteor events mostly observed below 90 km altitude with the Jicamarca VHF radar, but some events were also detected above 110 km. They called these events Low-Altitude Trail Echoes (LATE) because the majority of these events occurred below 90 km and did not show the time gap between the head-echo and the non-specular trails needed for plasma instabilities to manifest and grow. They noted that meteoroids with high mass could penetrate these low altitudes and hypothesize that fragmentation and high neutral density at these low altitudes should be considered in the characterization of these LATE events. Sugar et al. (2010) suggested that pockets of material with lower sublimation temperatures in the meteoroid could be the cause of these LATE events. They explained that once these pockets of meteoroid materials were released, plasma irregularities could rapidly develop in the trail due to a higher-density of plasma generated by these pockets. In contrast, we provide an alternative explanation based on the results reported in this paper. We suggest that it is possible for

neutral winds to be a mechanism that can influence the generation of LATE echoes. For example, Figure 9 shows a meteor simulation obtained using a 0.5 $\mu$g meteoroid traveling at 34 km/s. In this example, neutral wind shears affect the production of electrons in the final stage of the meteor trail, resulting in a trail echo spanning $\sim$1 km in range that has similar space-time signatures to LATE echoes. Notice that the creation of LATE events based on our findings requires an unusual combination of meteoroid occurrence and atmosphere background conditions, making LATE events rare. However, our results are not limited to the final stages of the meteoroid occurrence. We also expect to see LATE-like events at the initial stage of the meteoroid passage if the right conditions, such as background electron density, winds, etc., are satisfied. Nonetheless, low sublimation temperatures are also required to rapidly generate instabilities the proper combination of meteoroid and atmosphere properties would also create similar echoes at initial stages of the meteor trail as it was presented in Sugar et al. (2010).

## 4.2  Time-delay Between Head-echo and Non-specular Meteor Trail

Reported VHF radar meteor events typically show time-delays between head-echo and non-specular meteor trail that can vary from 0 s to $\sim$1 s (Mathews, 2004; Malhotra et al., 2007). Time-delays on the order of 20 to 30 ms have been associated with the time scales needed to generate plasma turbulence in the trail (Dyrud et al., 2002); while much longer time-delays values have been attributed to the time that is required to transport meteor plasma to regions where instabilities develop (Mathews, 2004). Also, Malhotra et al. (2007) has shown that the time-delay can also be a function of the viewing geometry between the radar **k**-vector and the Earth's magnetic field **B**. However, our meteor simulations reported in this paper present an additional scenario. For example, Figure 6 demonstrates that time-delays on the order of hundreds of milliseconds or longer can be generated when meteoroids have masses greater than 5 $\mu$g and produce dense non-specular trail that is affected by neutral winds with amplitudes close to the critical value, that are required, to sustain plasma instabilities. Our simulation results imply that instabilities slowly develop in these meteor trails because both polarization drifts and neutral winds do not have the strength to drive instabilities in the meteor trail rapidly. We should notice that diamagnetic drifts are divergence-free and therefore do not drive instabilities (Oppenheim et al., 2003).

## 5  Conclusions

This paper presented computer simulations of meteor trails over Salinas, Puerto Rico to understand the effects of neutral winds on meteor trail evolution. The non-specular meteor trail echo simulator (NSMES), which we described in this paper, is an important tool to study non-specular meteor trail echoes under different meteoroid masses and atmospheric conditions. It can be used to compare simulation results with experimental observations. For example, we showed using NSMES that it is plausible for meteor trails to exhibit unique radar signatures due to neutral winds that are useful to create a comprehensive characterization of meteor trails. We also discussed that time-delays between head-echo and non-specular trail on the order of hundreds of milliseconds or longer could be the result of dense trails that are affected by weak neutral wind amplitudes. Also, we provided experimental evidence that supported our simulation results and discussed their implications to other locations such as the Jicamarca VHF radar. It is not the purpose of this short paper to analyze all competing ideas in head-echo and

non-specular meteor trails research but to provide an alternative explanation in this research topic. We plan to fine-tune our meteor model and make the code open-source to the broader scientific community so others can verify our findings or expand our studies. We envision these efforts not to replace but rather complement more complex 3D meteor models (Oppenheim and Dimant, 2015; Dimant and Oppenheim, 2017a, b).

*Code and data availability.*  The data set for this paper is available online (http://www.datacommons.psu.edu/), following the data policy, for
details of this data and code, please contact J. Urbina at (jvu1@psu.edu).

*Author contributions.*  FG, JU, and LD conceived the project. FG did a significant part of the simulation and data analysis work. JU and LD helped with the discussion and interpretation of the results.

*Competing interests.*  The authors declare that they have no conflict of interest.

*Acknowledgements.*  This work is supported by the National Science Foundation under grants: ATM-0638624 and ATM-0457156 to Penn
State University. The authors would like to thank the JRO (and IGP) staff for performing the observations, and R. Sorbello for helping us in the preparation of this manuscript. The Jicamarca Radio Observatory is a facility of the Instituto Geofisico del Peru operated with support from the NSF AGS0905448 through Cornell University.

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

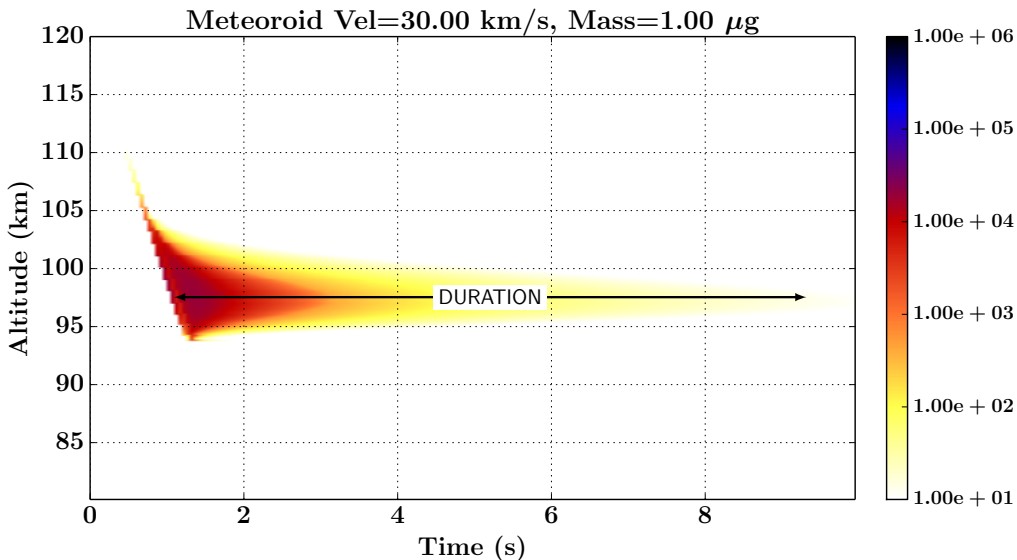

**Figure 1.** Simulated meteor head-echo and non-specular meteor trail echo. This simulation was obtained using a 1 $\mu$g meteoroid, traveling at 30 km/s at 45° entry angle. The duration of the non-specular echo is approximately 8 s. The color in the non-specular meteor trail echo indicates the plasma instability growth rate in s$^{-1}$, while for the simulated head-echo is the electron line density per meter divided by 10$^6$ (units chosen such that they appear on the same scale).

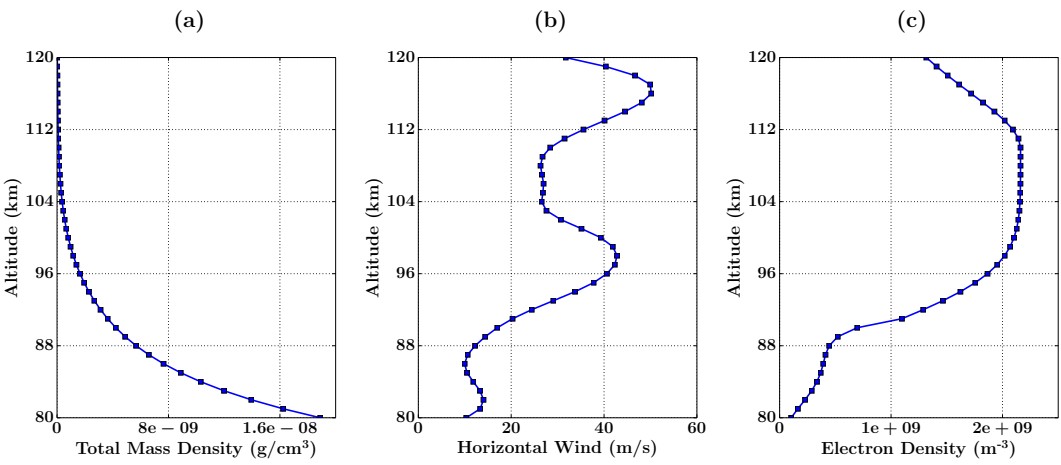

**Figure 2.** Corresponding atmospheric properties used to simulate the non-specular meteor echo shown in Figure 1; a) Background neutral density, b) Horizontal wind speeds, and c) Background electron density.

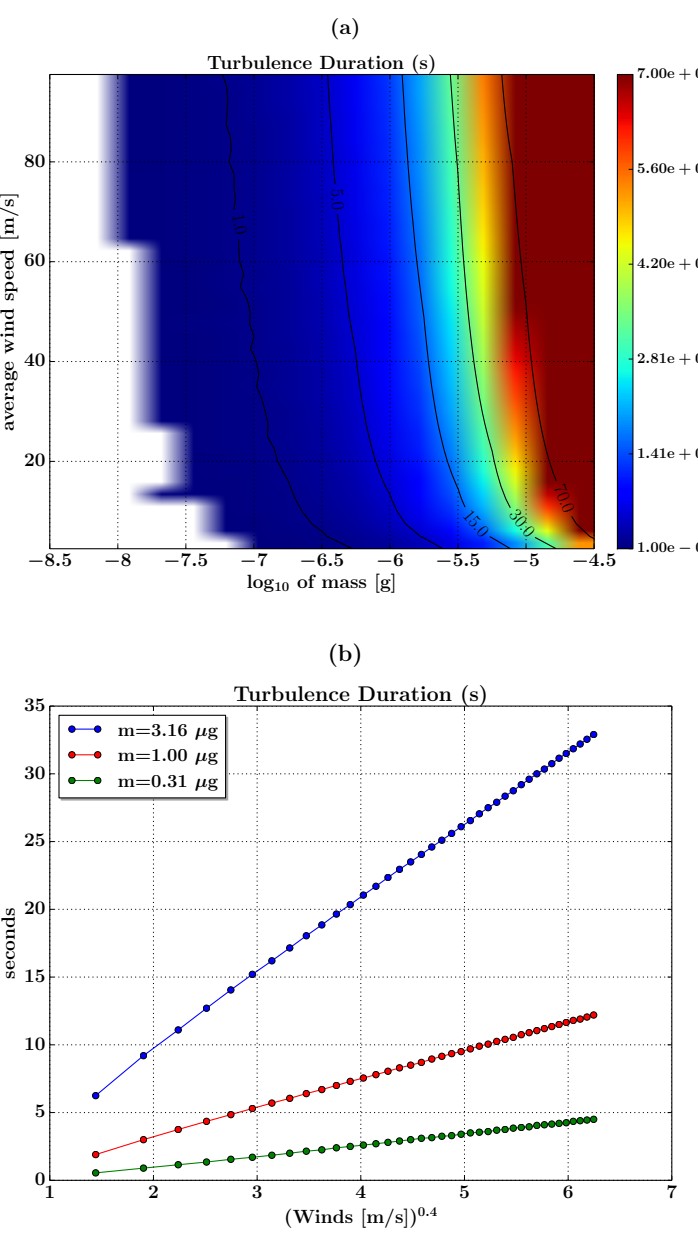

**Figure 3.** a) Meteor trail simulations to investigate the influence of neutral winds in trail duration. These simulations used a meteoroid speed of 35 km/s and masses between $3.16 \times 10^{-9}$-$3.16 \times 10^{-5}$g, and neutral wind amplitudes between 2.5 m/s and 100 m/s, b) Trail duration versus winds for three masses. Notice the linear relationship between duration and winds to the power of 0.4.

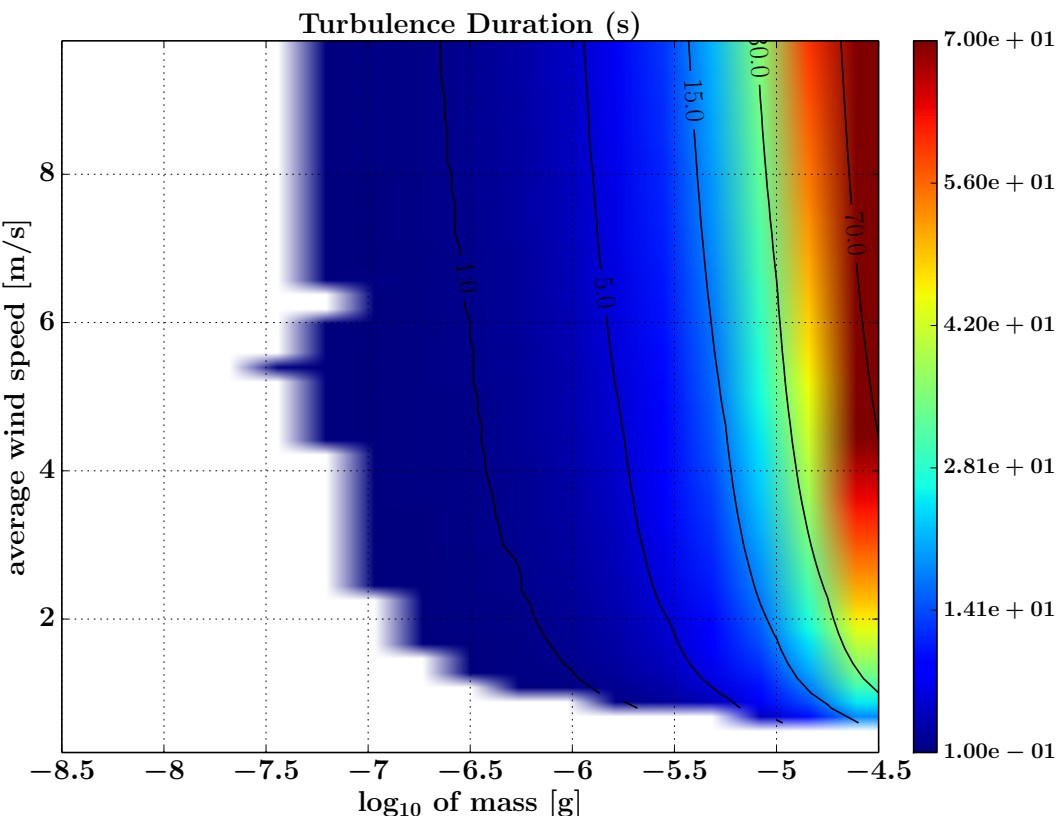

**Figure 4.** Minimum neutral winds vs. meteoroid mass. Notice that neutral winds smaller than 0.6 m/s do not create non-specular meteor trail echoes (i.e. plasma instabilities can not be sustained in the trail).

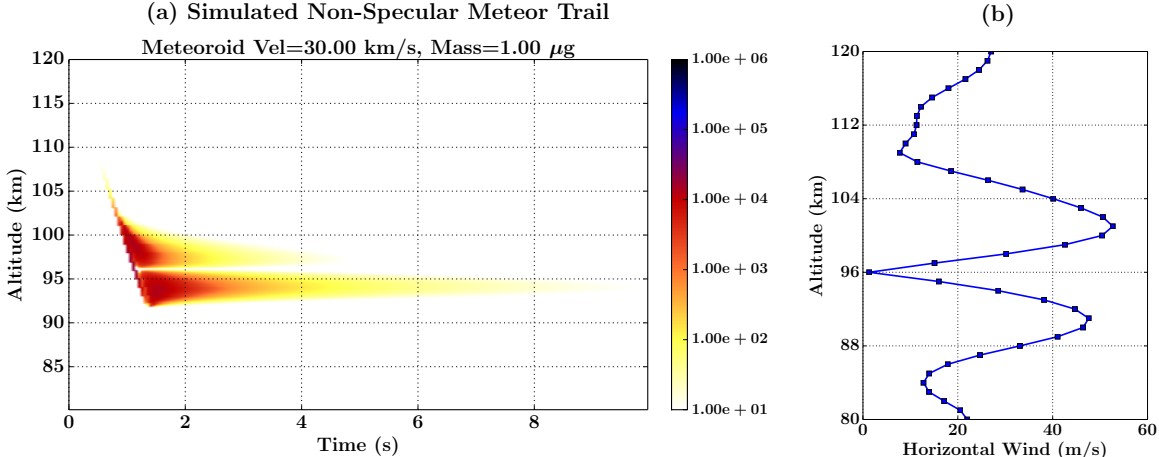

**Figure 5.** a) Simulated non-specular meteor trail using a 1 μg meteoroid traveling at 30 km/s, b) Corresponding neutral wind vertical profile. Notice strong wind shears near 96 km altitude. The absence of instabilities near 96 km is most probable due to the extremely low neutral winds at this altitude. Notice that neutral winds values are very small near 96 km but its magnitude rapidly increases below or above this altitude.

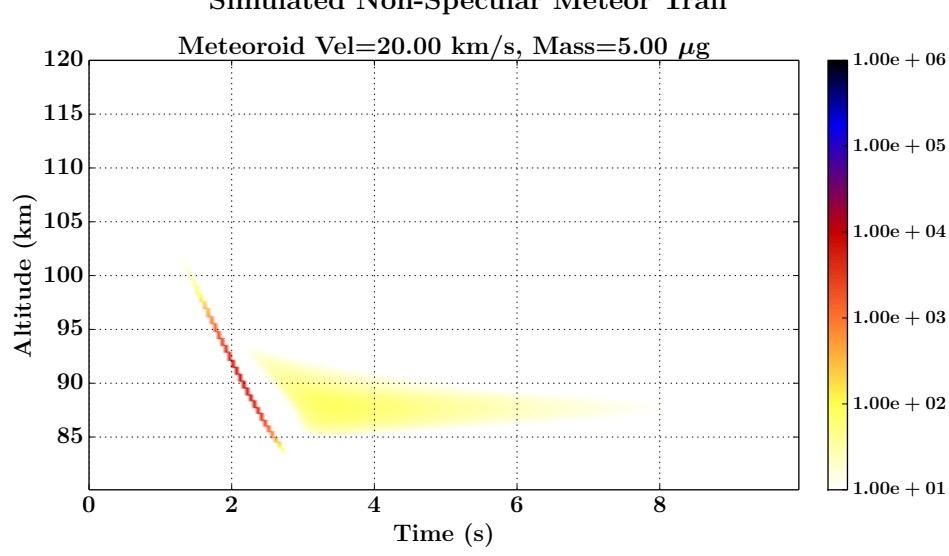

**Figure 6.** Meteor simulation to investigate the effect of weak neutral winds. In this example, instabilities take around 500 ms to develop due to neutral winds of 2.2 m/s.

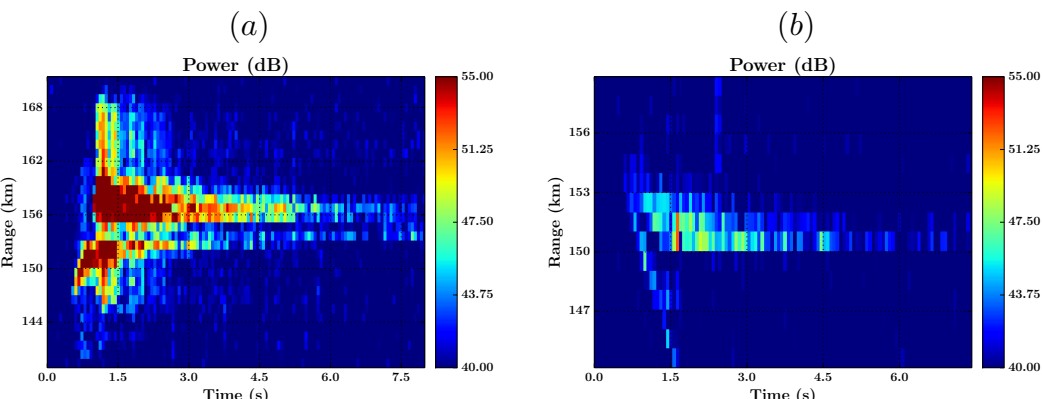

**Figure 7.** Meteor trails observed with the VHF radar operated in Salinas, Puerto Rico, during the Coqui-II Campaign. Panel (a) exhibits absence of backscattering power between 154 and 155 km range that resembles our simulation results shown in Figure 5, while Panel (b) shows that backscattering signal have a sudden decrease in ranges below 150 km.

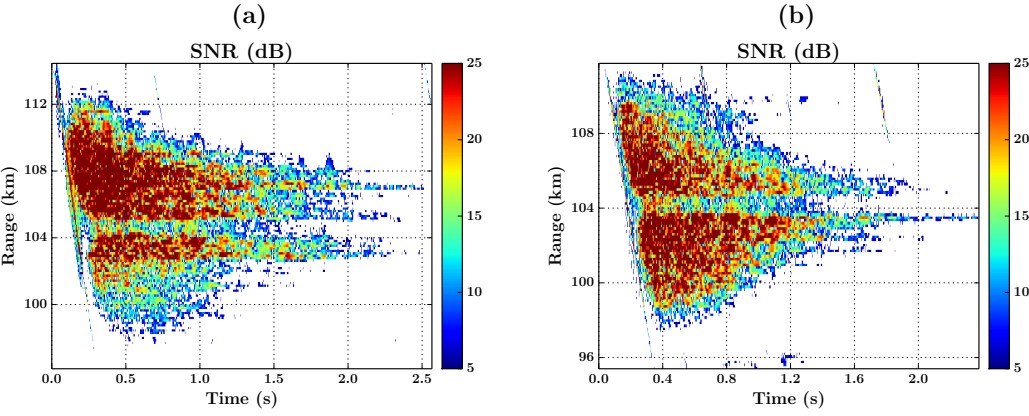

**Figure 8.** Panels (a) and (b) show meteor trails collected with Jicamarca VHF radar at 04:46 LT and 04:54 LT, respectively, on May 5th, 2007. The meteoroid in Panel (a) travels at 65 km/s, almost parallel, to the East-West magnetic direction, while the meteoroid in Panel (b) moves in the North-West magnetic direction, at speed of 56 km/s. Notice that both examples exhibit a sudden decrease in the backscatter power near 104.5 km altitude.

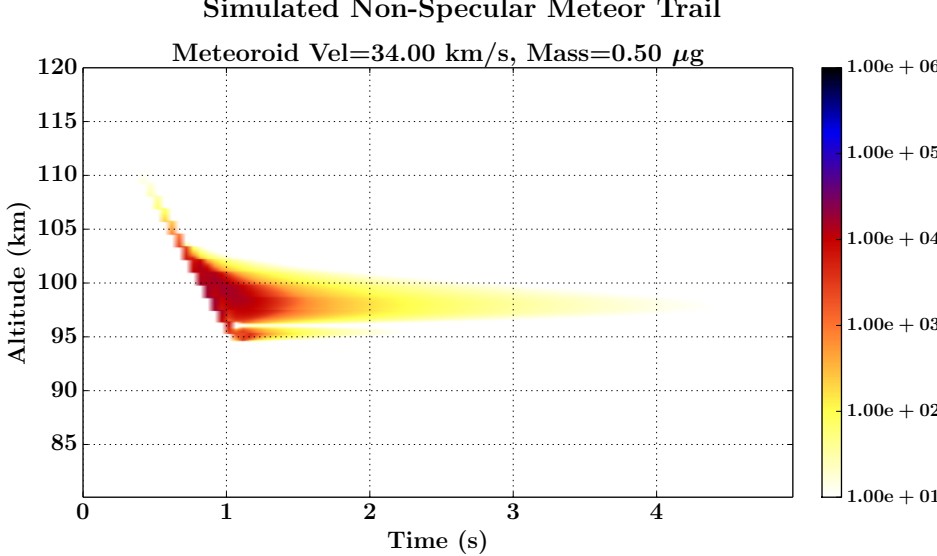

**Figure 9.** Simulation showing the effects of neutral wind shears, using a meteoroid with a mass of 0.5 $\mu$g that is traveling at 34 km/s. The background atmosphere/ionosphere conditions have a vertical profile similar to those displayed in Figure 5.