# Peer review of "Effect of neutral winds on the creation of non-specular meteor trail echoes"

_Annales Geophysicae, 2020_

## Referee Comment (RC1) · Anonymous Referee #1 · 9 Aug 2020

The manuscript discusses the effect of horizontal neutral wind on the generation and duration of nonspecular meteor echoes. The authors compare the simulation results from the nonspecular meteor trail echo simulator with radar observations, and report a threshold of horizontal neutral wind velocity 0.6 m/s controlling the generation of meteor trail irregularities. The results are important for understanding the nonspecular meteor echoes and associated meteor trail plasma instabilities. However, the physical process that how the horizontal wind affects or controls the generation of trail irregularities is not stated clearly and needs more clarification.

Specific comments,

lines 5-10: "for horizontal winds stronger than 1 m/s, a 0.316 $\mu$g meteoroid traveling at 35 km/s can produce meteor trail echo which is visible". Besides the properties of

meteor trail itself, the radar detection capability also determines whether the meteor trail is visible or not. In the comparison of the simulation results and observations, measurements from both small and large radars were used. It is not clear how the authors determine the visible/invisible meteor trail echo.

lines 101-103 and 160-162: Please explain in more detail how the horizontal winds produce and sustain plasma instabilities.

It is seen from Figure 5 that the trail echoes last the longest around 95 km altitude where the horizontal wind is small. Please explain.

lines 187-189 and Figure 8: Is it possible to derive the neutral wind from the nonspecular meteor echoes by using the method proposed by Oppenheim et al. (2009) and thus demonstrate the neutral wind shear? By using the meteor head echo, the meteoroid properties (e.g., mass, velocity) could also be derived. This provides a good chance to verify the simulation results.

---

## Referee Comment (RC2) · Anonymous Referee #2 · 3 Nov 2020

This manuscript presents an interesting analysis of radar observations of meteors and the effects of winds on these. It is trying to answer an important question in meteor physics: what are the effects of winds on wave growth in meteor trails and, hence, on non-specular radar detections. It uses a simple model developed more than 15 years ago and applies it to a range of atmospheric characteristics. However, as explained in the comments made below this manuscript leaves a lot of unanswered questions.

Major Comments:

1) Extensive observations at JRO (See Sugar et al, 2010 and Oppenheim et al, 2009) seem to show that, as trails span many km (sometimes over 15 km), through regions of the atmosphere with vastly different wind speeds trails seem to develop regardless of wind speed. This is odd because simulations and theory, like that presented in this

manuscript and more recent ones, imply one would expect that wave growth would depend on wind speed. This may result from drifts and currents that travel along the length of the trail, making the local wind speed less important to wave growth. 2) Line 50: The background has no references past 2011 but there has been extensive progress made on topics immediately relevant to the manuscript since then. 3) Line 55 Correct me if I'm wrong but this model is not exactly the state of the art and is a highly simplified 2-D model of a 3-D phenomenon. It may catch the basic physics but still, the authors should look into the more sophisticated models of instability for a 3-D meteor plasma of Dimant, et al (2015-2017). Also, the 3-D simulations of Oppenheim et. al (2015). 4) Line 167: These observations are interesting and help to make this case. However, when we've examined high-resolution images containing both head and trail echoes, we've generally seen that the heads and trails have gaps at the same ranges, implying that reduced returns were due to the nulls. Also, there are 2 papers where wind data was inferred from trail echoes and those winds go through zero and they still returned trail echoes. I agree that I would expect weaker trails when the winds are small but that is not what we have seen observationally. The field needs a larger statistical analysis of data comparing head and trail gaps to really see what the pattern is. A handful of cases will not be compelling either way because of the complexity and noise in this data. 5) Line 168: This image seems to show a gap in both head and trail, though the head isn't well resolved. Fig. 8a though does seem to show a reasonably strong head but a gap in the trail. This is intriguing but a single case is not sufficient. The Oppenheim, 2013 JGR shows that we do often get echoes at low velocities. 6) Line 219: The gap between the trail and the head is usually fairly constant or changing slowly, more so than the winds typically do.

Minor Comments: Line 19: "Past decade" -> at least two decades now (Chapin and Kudeki is over 25 years...)

Line 65: A summary of what physics is and is not in this model would be helpful to readers so they need not return to these 3 papers. I believe you could say it's a 2-D

local theory that assumes an infinite homogeneous trail and background (or something similar). It neglects physics along B or inhomogeneities of any kind. Or something similar.

Line 167: The word "considerable" is too vague.

Line 184: This implies this was a skimmer. Is that right? Line 186: "below or above" means all of them? And the feature referred to is unclear. Line 192: 11 years is not so recent. Line 204: These LATE flares also seem to effect the head echoes, are not all at low altitudes and are quite rare (while wind shears are not). Line 214: Divergence not "divergent"

---

## Author Comment (AC1) · 13 Jan 2021

**Authors' response to the comments of reviewer 1:**

We thank the reviewer for the suggestions and comments. Below, we present our responses to each of those comments. We have repeated the relevant comments for convenience in each response and then provide the text (in blue) we intend to add in the revised manuscript.

**General Comments:** The manuscript discusses the effect of horizontal neutral wind on the generation and duration of nonspecular meteor echoes. The authors compare the simulation results from the nonspecular meteor trail echo simulator with radar observations, and report a threshold of horizontal neutral wind velocity 0.6 m/s controlling the

generation of meteor trail irregularities. The results are important for understanding the nonspecular meteor echoes and associated meteor trail plasma instabilities. However, the physical process that how the horizontal wind affects or controls the generation of trail irregularities is not stated clearly and needs more clarification.

**Response:** The physics of the model is described in several papers: Dyrud et al. (2002), Oppenheim et al. (2001), Oppenheim et al. (2003), Dyrud et al. (2005), and Dyrud et al. (2007). In the revised version, we will add a summary of what physics in this model, so readers do not need to return to these papers. We will modify the text between lines 55-60 to provide a summary of meteor physics included in the model as follows.

As explained in these papers, the model starts by computing the amount of ablated particles created behind the meteoroid body. These energetic particles are then used to calculate the amount of ionization made in the trail. Here we assume that the ionization created in the trail is initially distributed in a cylindrical volume defined by the initial radius. At this point, the trail is expanded by either ambipolar diffusion or turbulent diffusion to simulate the absence or presence of plasma instabilities in the trail during its evolution (Dyrud et al., 2001; Yee and Close, 2013). The plasma instability analysis is based on meteor Farley-Buneman Gradient-Drift (FBGD) instability reported in Dyrud et al. (2002), Oppenheim et al. (2001), and Oppenheim et al. (2003). NSMES assumes that a non-specular meteor trail echo is created because the trail becomes Bragg reflective at altitudes, where plasma instabilities can develop (Dyrud et al., 2002). The simulations produce artificial radar range-time-intensity (RTI) images that we use as proxies to help us in the analysis of Coqui-II and Jicamarca meteor observations.

**Specific comments**

**lines 5-10:** "for horizontal winds stronger than 1 m/s, a 0.316 $\mu$g meteoroid traveling at 35 km/s can produce meteor trail echo which is visible". Besides the properties of
meteor trail itself, the radar detection capability also determines whether the meteor trail is visible or not. In the comparison of the simulation results and observations, measurements from both small and large radars were used. It is not clear how the authors determine the visible/invisible meteor trail echo.

**Response:** This paper focuses on simulations of meteor trail echoes due to the meteoroid properties and the background atmosphere. The neutral velocity threshold illustrates how simulations show that no trail echo is created below a critical wind value. We agree with the reviewer that any radar echo is a function of both the instrument transfer function and the physics of the targets it is probing in practice. So, the critical wind value can't be mapped directly to radar observations but can be used to shed light on the physics of meteor trails and improve their modeling. We will add the paragraph below near line 10 to make it clear what we mean by visible/invisible meteor trails echo.

The neutral velocity threshold illustrates how simulations show that no trail echo is created below a critical wind value. This critical wind value is not mapped directly to radar observations but it is used to shed light on the physics of meteor trails and improve their modeling.

**lines 101-103 and 160-162:** Please explain in more detail how the horizontal winds produce and sustain plasma instabilities.

**Response:** As we indicated in the response to general comments, the physics of the model is described in several papers. We will add these references near line 103 to clarify this comment.

Dyrud et al. (2002), Oppenheim et al. (2001), and Oppenheim et al. (2003).

And add these references near 162.

Dyrud et al. (2002), Oppenheim et al. (2001).

**It is seen from Figure 5 that the trail echoes last the longest around 95 km altitude where the horizontal wind is small. Please explain.**

**Response:** Notice that neutral winds values are very small near 96 km but its magnitude rapidly increases below this altitude so near 95 km neutral winds exhibit values larger than 4 m/s. We point out this effect in a broader sense in Figure 4. We will add this clarification in the manuscript near line 127.

**lines 187-189 and Figure 8:** Is it possible to derive the neutral wind from the non-specular meteor echoes by using the method proposed by Oppenheim et al. (2009) and thus demonstrate the neutral wind shear? By using the meteor head echo, the meteoroid properties (e.g., mass, velocity) could also be derived. This provides a good chance to verify the simulation results.

**Response:** This is an excellent observation. That would be the next step. It isn't a trivial problem since neutral winds estimates proposed in Oppenheim et al. (2009) need to be validated with other techniques first. We indicated that that is our intention in lines 189-191 of the manuscript.

---

## Author Comment (AC2) · 13 Jan 2021

**Authors' response to the comments of reviewer 2:**

We thank the reviewer for the suggestions and comments. Below, we present our responses to each of those comments. We have repeated the relevant comments for convenience in each response and then provide the text (in blue) we intend to add in the revised manuscript.

**General Comments:** This manuscript presents an interesting analysis of radar observations of meteors and the effects of winds on these. It is trying to answer an important question in meteor physics: what are the effects of winds on wave growth in meteor trails and, hence, on non-specular radar detections. It uses a simple model devel-
oped more than 15 years ago and applies it to a range of atmospheric characteristics. However, as explained in the comments made below this manuscript leaves a lot of unanswered questions.

**Response:** Below are the points we want to address in this study. We will add these points to the revised version of the manuscript.

- Our model and the associated software can be executed in a general-purpose PC-based system. It can easily be adapted and combined with other tools to study very large meteor populations. In contrast, as far as we know, more sophisticated 3D meteor models require supercomputer clusters and do not fully simulate the actual extent of a meteor trail or produce results that can be closely compared to 2D observational data.

- Although our numerical model is a simplified representation of the meteor physics, it can produce very good and fine details such as those reported in this paper. Our model can be used to account for and understand the statistical outcome of thousands of meteors acting collectively on the Earth's Upper Atmosphere.

- We plan to fine-tune our meteor model and plan to make the code open-source to the scientific community so others can verify our findings or expand our results. We envision our efforts not to replace but complement more complex 3D meteor models.

**Major Comments: 1)** Extensive observations at JRO (See Sugar et al, 2010 and Oppenheim et al, 2009) seem to show that, as trails span many km (sometimes over 15 km), through regions of the atmosphere with vastly different wind speeds trails seem to develop regardless of wind speed. This is odd because simulations and theory, like that presented in this manuscript and more recent ones, imply one would expect that wave growth would depend on wind speed. This may result from drifts and currents
that travel along the length of the trail, making the local wind speed less important to wave growth.

**Response:** This is a very important observation, and we partially agree with the reviewer. We are developing a deep learning algorithm to detect and classify thousands of meteors properly. Once this algorithm is fully functional, we plan to expand the results reported in this manuscript and perform an extensive statistical analysis of trail echoes with gaps.

**2) Line 50:** The background has no references past 2011 but there has been extensive progress made on topics immediately relevant to the manuscript since then.

**Response:** We will add these additional references:

1. Oppenheim, M. M., S. Arredondo, and G. Sugar (2014), Intense winds and shears in the equatorial lower thermosphere measured by high-resolution nonspecular meteor radar, J. Geophys. Res. Space Physics, 119, 2178–2186, doi:10.1002/2013JA019272.

2. Oppenheim, M. M., and Y. S. Dimant(2015), First 3-D simulations of meteor plasma dynamics and turbulence, Geophys. Res. Lett., 42, 681–687, doi:10.1002/2014GL062411

3. Dimant, Y. S. and M. M. Oppenheim(2017), Formation of plasma arounda small meteoroid: 1. Kinetic theory,J. Geophys. Res. Space Physics,122, 4669–4696,doi:10.1002/2017JA023960.

4. Chau, J. L., Strelnikova, I., Schult, C., Oppenheim, M. M., Kelley, M. C., Stober, G., and Singer, W. (2014), Nonspecular meteor trails from non‐field‐aligned irregularities: Can they be explained by presence of charged meteor dust?, Geophys. Res. Lett., 41, 3336– 3343, doi:10.1002/2014GL059922..

5. Dimant, Y. S., and Oppenheim, M. M. (2017), Formation of plasma around a small meteoroid: 2. Implications for radar head echo, J. Geophys. Res. Space Physics, 122, 4697– 4711, doi:10.1002/2017JA023963.

6. Sugar, G., Oppenheim, M. M., Dimant, Y. S., Close, S. (2018). Formation of plasma around a small meteoroid: Simulation and theory. Journal of Geophysical Research: Space Physics, 123, 4080– 4093. https://doi.org/10.1002/2018JA025265

7. Sugar, G., Oppenheim, M. M., Dimant, Y. S., Close, S. (2019). Formation of plasma around a small meteoroid: Electrostatic simulations. Journal of Geophysical Research: Space Physics, 124, 3810– 3826. https://doi.org/10.1029/2018JA026434

**3) Line 55** Correct me if I'm wrong but this model is not exactly the state of the art and is a highly simplified 2-D model of a 3-D phenomenon. It may catch the basic physics but still, the authors should look into the more sophisticated models of instability for a 3-D meteor plasma of Dimant, et al (2015-2017). Also, the 3-D simulations of Oppenheim et. Al (2015).

**Response:** Rather than attempting to reproduce these more complex 3D simulations pointed out by the reviewer, our research efforts are complementary to this comment. In the response to the general comments, we provided the main points we seek to address with this study. We will include these relevant references pointed out by the reviewer in the revised version of the manuscript.

**4) Line 167:** These observations are interesting and help to make this case. However, when we've examined high-resolution images containing both head and trail echoes, we've generally seen that the heads and trails have gaps at the same ranges, implying that reduced returns were due to the nulls. Also, there are 2 papers where wind data was inferred from trail echoes and those winds go through zero and they still returned trail echoes. I agree that I would expect weaker trails when the winds are small but that is not what we have seen observationally. The field needs a larger statistical analysis of data comparing head and trail gaps to really see what the pattern is. A handful of cases will not be compelling either way because of the complexity and noise in this data.

**Response:** We agree with the reviewer that antenna nulls and noise levels in the data

could be reasonable explanations to account for certain meteor events when gaps are observed in both the head and trail echoes at the same ranges. We are developing a deep learning algorithm to detect and classify automatically thousands of meteors properly near real-time. Once this algorithm is fully functional, we plan to expand the results reported in this manuscript and perform an extensive statistical analysis of trail echoes with gaps. We also expect to carry out future radar experiments and compute neutral wind amplitudes using meteor trails as described in (Oppenheim et al., 2009) to establish a complete understanding of the gaps shown in this paper. We will add this text near line 167 to address some of these comments.

Routine meteor interferometry analysis, described in (Chau and Galindo, 2008), was applied to the events presented in this paper. We analyzed each of the head echo events using both received SNR and interferometry analysis. We discarded noise level as a potential explanation since in the examples we report, all trail echoes were at least 3 dB above the noise level. So statistically, it is improbable that noise is responsible for gaps in different echoes at the same range and around the same time. We also discarded antenna nulls as a possible explanation for the trails' gaps since interferometry analysis placed these events in the main lobe of the antenna. These examples from Jicamarca have echo gaps observed only in the trail echoes, as shown in Figure 8a. Notice that there is no drop in power intensity for the head echo around the 104.5 km range when zoom-in in this figure.

**5) Line 168:** This image seems to show a gap in both head and trail, though the head isn't well resolved. Fig. 8a though does seem to show a reasonably strong head but a gap in the trail. This is intriguing but a single case is not sufficient. The Oppenheim, 2013 JGR shows that we do often get echoes at low velocities.

**Response:** Figure 7 shows two examples collected with a medium power VHF radar (Urbina et al., 2000), while Figure 8 displays two events probed with Jicamarca HPLA radar. Like we indicate in the manuscript near line 180 that we found 17 (out of 103). These events were manually classified and analyzed. Since this approach is quite

tedious, we are currently developing a deep learning algorithm to detect and classify thousands of meteors correctly in almost real-time. Once this algorithm is fully functional, we plan to expand the results reported in this manuscript and perform an extensive statistical analysis of trail echoes with gaps.

**6) Line 219:** The gap between the trail and the head is usually fairly constant or changing slowly, more so than the winds typically do.

**Response:** We agree with the reviewer. This is why we showed this simulation to illustrate how low values neutral winds can produce these head-trails pairs with larger gaps. In practice, detecting a trail echo also depends on other factors such as the radar transfer function, receiver bandwidth, noise, etc.

**Minor Comments:**

**Line 19:** "Past decade" -> at least two decades now (Chapin and Kudeki is over 25 years...)

**Response:** We have changed this expression to For more than two decades.

**Line 65:** A summary of what physics is and is not in this model would be helpful to readers so they need not return to these 3 papers. I believe you could say it's a 2-D local theory that assumes an infinite homogeneous trail and background (or something similar). It neglects physics along B or inhomogeneities of any kind. Or something similar.

**Response:** We will modify the text between lines 55-60 to provide a summary of meteor physics included in the model as follows.

As explained in these papers, the model starts by computing the amount of ablated particles created behind the meteoroid body. These energetic particles are then used to calculate the amount of ionization made in the trail. Here we assume that the ionization created in the trail is initially distributed in a cylindrical volume defined by the initial radius. At this point, the trail is expanded by either ambipolar diffusion or turbulent diffusion to simulate the absence or presence of plasma instabilities in the trail during its evolution (Dyrud et al., 2001; Yee and Close, 2013). The plasma instability analysis is based on meteor Farley-Buneman Gradient-Drift (FBGD) instability reported in Dyrud et al. (2002), Oppenheim et al. (2001), and Oppenheim et al. (2003). NSMES assumes that a non-specular meteor trail echo is created because the trail becomes Bragg reflective at altitudes, where plasma instabilities can develop (Dyrud et al., 2002). The simulations produce artificial radar range-time-intensity (RTI) images that we use as proxies to help us in the analysis of Coqui-II and Jicamarca meteor observations.

**Line 167:** The word "considerable" is too vague.

**Response:** We have replaced this term with more than 1000

**Line 184:** This implies this was a skimmer. Is that right?

**Response:** Both meteor events shown in Figure 8 exhibit elevation angles around 70 degrees and appear to traverse the main beam.

**Line 186:** "below or above" means all of them? And the feature referred to is unclear.

**Response:** Yes, we meant the remaining meteors observed around this period of time do not exhibit the feature under discussion. We will remove the sentence since the authors believe it isn't necessary and is confusing the reader.

We will delete: "Meteor trails observed below or above 104.5 km altitude do not exhibit this feature."

**Line 192:** 11 years is not so recent.

**Response:** We will remove the term "recent paper."

**Line 204:** These LATE flares also seem to effect the head echoes, are not all at low altitudes and are quite rare (while wind shears are not).

**Response:** We agree that wind shears aren't rare. However, the proper combination of physical parameters to create a FLARE event based on our findings represent a rare situation. We will clarify these comments in the paper by editing lines 204-207 as follows:

Notice that the creation of LATE events based on our findings requires an unusual combination of meteoroid occurrence and atmosphere background conditions, making LATE events rare. However, our results are not only limited to the final stages of the meteoroid occurrence. We also expect to see LATE-like events at the initial stage of the meteoroid passage if the right conditions, such as background electron density, winds, etc., are satisfied.

**Line 214:** Divergence not "divergent"

**Response:** We agree, it should be "divergence."

---

## Author Response (AR1)

**Authors' response to the comments of reviewer 1:**

We thank the reviewer for the suggestions and comments. Below, we present our responses to each of those comments. The text (in red) has been added to the revised manuscript. The line number indicated in the responses below is with respect to the marked-up version of the revised manuscript using track changes generate using *latexdiff* in LaTeX.

**General Comments:** The manuscript discusses the effect of horizontal neutral wind on the generation and duration of non-specular meteor echoes. The authors compare the simulation results from the non-specular meteor trail echo simulator with radar observations, and report a threshold of horizontal neutral wind velocity 0.6 m/s controlling the generation of meteor trail irregularities. The results are important for understanding the non-specular meteor echoes and associated meteor trail plasma instabilities. However, the physical process that how the horizontal wind affects or controls the generation of trail irregularities is not stated clearly and needs more clarification.

**Response:** The physics of the model is described in several papers: Dyrud et al. (2002), Oppenheim et al. (2001), Oppenheim et al. (2003), Dyrud et al. (2005), and Dyrud et al. (2007). We have added the text below between lines 60-65 to provide a summary of meteor physics discussed in these papers.

As explained in these papers, the model starts by computing the amount of ablated particles created behind the meteoroid body. These energetic particles are then used to calculate the amount of ionization made in the trail. Here we assume that the ionization created in the trail is initially distributed in a cylindrical volume defined by the initial radius. At this point, the trail is expanded by either ambipolar diffusion or turbulent diffusion to simulate the absence or presence of plasma instabilities in the trail during its evolution (Dyrud et al., 2001; Yee and Close, 2013).

**Specific comments**
**lines 5-10:** "for horizontal winds stronger than 1 m/s, a 0.316 µg meteoroid traveling at 35 km/s can produce meteor trail echo which is visible". Besides the properties of meteor trail itself, the radar detection capability also determines whether the meteor trail is visible or not. In the comparison of the simulation results and observations, measurements from both small and large radars were used. It is not clear how the authors determine the visible/invisible meteor trail echo.

**Response:** This paper focuses on simulations of meteor trail echoes due to the meteoroid properties and the background atmosphere. The neutral velocity threshold illustrates how simulations show that no trail echo is created below a critical wind value. We agree with the reviewer that any radar echo is a function of both the instrument transfer function and the physics of the targets it is probing in practice. So, the critical wind value can't be mapped directly to radar observations but can be used to shed light on the physics of meteor trails and improve their modeling. We added the paragraph below near line 10 to make it clear what we mean by visible/invisible meteor trails echo.

The neutral velocity threshold illustrates how simulations show that no trail echo is created below a critical wind value. This critical wind value is not mapped directly to radar observations but it is used to shed light on the physics of meteor trails and improve their modeling.

**lines 101-103 and 160-162:** Please explain in more detail how the horizontal winds produce and sustain plasma instabilities.

**Response:** As we indicated in the response to general comments, the physics of the model is described in several papers. We have added the text and references below near line 115 to clarify this comment.

Additional details about the meteor physics can be found in (Oppenheim et al., 2001; Dyrud et al., 2002; and Oppenheim et al. (2003)).

**It is seen from Figure 5 that the trail echoes last the longest around 95 km altitude where the horizontal wind is small. Please explain.**

**Response:** Notice that neutral winds values are very small near 96 km but its magnitude rapidly increases below this altitude so near 95 km neutral winds exhibit values larger than 4 m/s. We point out this effect in a broader sense in Figure 4.

We have added this text near line 153 to address this concern.

Notice that neutral winds values are very small near 96 km but their magnitude rapidly increases below or above this altitude to sustain plasma instabilities.

**lines 187-189 and Figure 8:** Is it possible to derive the neutral wind from the non- specular meteor echoes by using the method proposed by Oppenheim et al. (2009) and thus demonstrate the neutral wind shear? By using the meteor head echo, the meteoroid properties (e.g., mass, velocity) could also be derived. This provides a good chance to verify the simulation results.

**Response:** This is an excellent observation. That would be the next step. It isn't a trivial problem since neutral winds estimates proposed in Oppenheim et al. (2009) need to be validated with other techniques first. We indicated that that is our intention in lines 211-212 of the manuscript.

**Authors' response to the comments of reviewer 2:**

We thank the reviewer for the suggestions and comments. Below, we present our responses to each of those comments. The text (in red) has been added to the revised manuscript. The line number indicated in the responses below is with respect to the marked-up version of the revised manuscript using track changes generate using *latexdiff* in LaTeX.

**General Comments:** This manuscript presents an interesting analysis of radar observations of meteors and the effects of winds on these. It is trying to answer an important question in meteor physics: what are the effects of winds on wave growth in meteor trails and, hence, on non-specular radar detections. It uses a simple model developed more than 15 years ago and applies it to a range of atmospheric characteristics. However, as explained in the comments made below this manuscript leaves a lot of unanswered questions.

**Response:** Below are the points we want to address in this study. In lines 73-79 of the revised manuscript, we added this text:

Our model and the associated software can be executed in a general-purpose PC-based system. It can easily be adapted and combined with other tools to study very large meteor populations. In contrast, as far

as we know, more sophisticated 3D meteor models require supercomputer clusters and do not fully simulate the actual extent of a meteor trail or produce results that can be closely compared to 2D observational data. Although our numerical model is a simplified representation of the meteor physics, it can produce very good and fine details such as those reported in this paper. Our model can be used to account for and understand the statistical outcome of thousands of meteors acting collectively on the Earth's Upper Atmosphere.

In lines 256-259, we added this text:

We plan to fine-tune our meteor model and plan to make the code open-source to the scientific community so others can verify our findings or expand our results. We envision our efforts not to replace but complement more complex 3D meteor models.

**Major Comments:**

**1)** Extensive observations at JRO (See Sugar et al, 2010 and Oppenheim et al, 2009) seem to show that, as trails span many km (sometimes over 15 km), through regions of the atmosphere with vastly different wind speeds trails seem to develop regardless of wind speed. This is odd because simulations and theory, like that presented in this manuscript and more recent ones, imply one would expect that wave growth would depend on wind speed. This may result from drifts and currents that travel along the length of the trail, making the local wind speed less important to wave growth.

**Response:** This is a very important observation, and we partially agree with the reviewer. We are developing a deep learning algorithm to carefully detect and classify thousands of meteors near real-time. Once this algorithm is fully validated and functional, we plan to expand the results reported in this manuscript and perform an extensive statistical analysis of trail echoes with gaps.

**2) Line 50:** The background has no references past 2011 but there has been extensive progress made on topics immediately relevant to the manuscript since then.

**Response:** We have added these additional references near line 25:

1. Oppenheim, M. M., S. Arredondo, and G. Sugar (2014), Intense winds and shears in the equatorial lower thermosphere measured by high-resolution nonspecular meteor radar, J. Geophys. Res. Space Physics, 119, 2178–2186, doi:10.1002/2013JA019272.

2. Oppenheim, M. M., and Y. S. Dimant(2015), First 3-D simulations of me- teor plasma dynamics and turbulence, Geophys. Res. Lett., 42, 681–687, doi:10.1002/2014GL062411

3. Dimant, Y. S. and M. M. Oppenheim(2017), Formation of plasma arounda small meteoroid: 1. Kinetic theory,J. Geophys. Res. Space Physics,122, 4669– 4696,doi:10.1002/2017JA023960.

4. Chau, J. L., Strelnikova, I., Schult, C., Oppenheim, M. M., Kelley, M. C., Stober, G., and Singer, W. (2014), Nonspecular meteor trails from non field aligned irregularities: Can they be explained by presence of charged meteor dust?, Geophys. Res. Lett., 41, 3336– 3343, doi:10.1002/2014GL059922..

5. Dimant, Y. S., and Oppenheim, M. M. (2017), Formation of plasma around a small meteoroid: 2. Implications for radar head echo, J. Geophys. Res. Space Physics, 122, 4697– 4711, doi:10.1002/2017JA023963.

6. Sugar, G., Oppenheim, M. M., Dimant, Y. S., Close, S. (2018). Formation of plasma around a small meteoroid: Simulation and theory. Journal of Geophysical Research: Space Physics, 123, 4080– 4093. https://doi.org/10.1002/2018JA025265

7. Sugar, G., Oppenheim, M. M., Dimant, Y. S., Close, S. (2019). Formation of plasma around a small meteoroid: Electrostatic simulations. Journal of Geophysical Research: Space Physics, 124, 3810– 3826. https://doi.org/10.1029/2018JA026434

**3) Line 55** Correct me if I'm wrong but this model is not exactly the state of the art and is a highly simplified 2-D model of a 3-D phenomenon. It may catch the basic physics but still, the authors should look into the more sophisticated models of instability for a 3-D meteor plasma of Dimant, et al (2015-2017). Also, the 3-D simulations of Oppenheim et. Al (2015).

**Response:** Rather than attempting to reproduce these more complex 3D simulations pointed out by the reviewer, our research efforts are complementary to this comment. In the response to the general comments, we provided the main points (and text lines) we seek to address with this study.

**4) Line 167:** These observations are interesting and help to make this case. However, when we've examined high-resolution images containing both head and trail echoes, we've generally seen that the heads and trails have gaps at the same ranges, implying that reduced returns were due to the nulls. Also, there are 2 papers where wind data was inferred from trail echoes and those winds go through zero and they still returned trail echoes. I agree that I would expect weaker trails when the winds are small but that is not what we have seen observationally. The field needs a larger statistical analysis of data comparing head and trail gaps to really see what the pattern is. A handful of cases will not be compelling either way because of the complexity and noise in this data.

**Response:** We agree with the reviewer that antenna nulls and noise levels in the data could be reasonable explanations to account for certain meteor events when gaps are observed in both the head and trail echoes at the same ranges. As we indicated earlier, we are developing a deep learning algorithm to automatically detect and classify thousands of meteors near real-time. Once this algorithm is fully operational, we plan to expand the results reported in this manuscript and perform an extensive statistical analysis of trail echoes with gaps. We also expect to carry out future radar experiments and compute neutral wind amplitudes using meteor trails as described in (Oppenheim et al., 2009) to establish a complete understanding of the gaps shown in this paper. We have added the text below near line 199 to address these comments.

We analyzed each of the head echo events using both received SNR and interferometry analysis. We discarded noise level as a potential explanation since in the examples we report, all trail echoes were at least 3 dB above the noise level. So statistically, it is improbable that noise is responsible for gaps in different echoes at the same range and around the same time. We also discarded antenna nulls as a possible explanation for the trails' gaps since interferometry analysis placed these events in the main lobe of the antenna. These examples from Jicamarca have echo gaps observed only in the trail echoes, as shown in Figure 8a. Notice that there is no drop in power intensity for the head echo around the 104.5 km range when zoom-in in this figure.

**5) Line 168:** This image seems to show a gap in both head and trail, though the head isn't well resolved. Fig. 8a though does seem to show a reasonably strong head but a gap in the trail. This is intriguing but a

single case is not sufficient. The Oppenheim, 2013 JGR shows that we do often get echoes at low velocities.

**Response:** Figure 7 shows two examples collected with a medium power VHF radar (Urbina et al., 2000), while Figure 8 displays two events probed with Jicamarca HPLA radar. Like we indicate in the manuscript near line 196 that we found 17 (out of 103). These events were manually classified and analyzed. Since this approach is quite tedious and time-consuming, we are currently developing a deep learning algorithm to correctly detect and classify thousands of meteors. Our goal is to make this deep learning algorithm easily portable to other HPLA radars around the world. Once this algorithm has been validated, we plan to expand the results reported in this manuscript and perform an extensive statistical analysis of trail echoes with gaps.

**6) Line 219:** The gap between the trail and the head is usually fairly constant or changing slowly, more so than the winds typically do.

**Response:** We agree with the reviewer. This is why we showed this simulation to illustrate how low values neutral winds can produce these head-trails pairs with larger gaps. In practice, detecting a trail echo also depends on other factors such as the radar transfer function, which includes RF-frontends, receiver bandwidth, filters, antenna geometry, noise, etc.

**Minor Comments:**
**Line 19:** "Past decade" -> at least two decades now (Chapin and Kudeki is over 25 years...)

**Response:** In line 21, we have changed this expression to: For more than two decades

**Line 65:** A summary of what physics is and is not in this model would be helpful to readers so they need not return to these 3 papers. I believe you could say it's a 2-D local theory that assumes an infinite homogeneous trail and background (or something similar). It neglects physics along B or inhomogeneities of any kind. Or something similar.

**Response:** We have added the text below between lines 58-64 to provide a summary of meteor physics included in the model as follows.

As explained in these papers, the model starts by computing the amount of ablated particles created behind the meteoroid body. These energetic particles are then used to calculate the amount of ionization made in the trail. Here we assume that the ionization created in the trail is initially distributed in a cylindrical volume defined by the initial radius. At this point, the trail is expanded by either ambipolar diffusion or turbulent diffusion to simulate the absence or presence of plasma instabilities in the trail during its evolution (Dyrud et al., 2001; Yee and Close, 2013).

**Line 167:** The word "considerable" is too vague.

**Response:** In line 186, we have replaced this term with more than 1000

**Line 184:** This implies this was a skimmer. Is that right?

**Response:** Both meteor events shown in Figure 8 exhibit elevation angles around 70 degrees and appear to traverse the main beam.

**Line 186:** "below or above" means all of them? And the feature referred to is unclear.

**Response:** Yes, we meant the remaining meteors observed around this period of time do not exhibit the feature under discussion. We deleted the sentence since the authors believe it isn't necessary and is confusing the reader.

We deleted: Meteor trails observed below or above 104.5 km altitude do not exhibit this feature.

**Line 192:** 11 years is not so recent.
**Response:** In line 214, we deleted the term: recent paper

**Line 204:** These LATE flares also seem to effect the head echoes, are not all at low altitudes and are quite rare (while wind shears are not).

**Response:** We agree that wind shears aren't rare. However, the proper combination of physical parameters to create a FLARE event based on our findings represent a rare situation. We clarified these comments in the paper by editing lines 226-230 as follows:

Notice that the creation of LATE events based on our findings requires an unusual combination of meteoroid occurrence and atmosphere background conditions, making LATE events rare. However, our results are not only limited to the final stages of the meteoroid occurrence. We also expect to see LATE-like events at the initial stage of the meteoroid passage if the right conditions, such as background electron density, winds, etc., are satisfied. Nonetheless,

**Line 214:** Divergence not "divergent" **Response:** We agree, it should be: divergence

---

## Author Response (AR2)

Dr. Keisuke Hosokawa
Editor
Annales Geophysicae

Dear Dr. Hosokawa:

Please find attached a revised version of our manuscript, "Effect of neutral winds on the creation of non-specular meteor trail echoes", which we are submitting for your consideration for publication in Annales Geophysicae. The text (in blue) has been added to the revised manuscript. The line number indicated in the responses below is with respect to the marked-up version of the revised manuscript using track changes generate using *latexdiff* in LaTeX.

Our revision to address the issue raised by the reviewer is listed below. We hope these responses are useful in your final evaluation.

Thank you for considering our manuscript.

Sincerely,
Julio Urbina

**Suggestions for revision or reasons for rejection (will be published if the paper is accepted for final publication)**

---> lines 187-189 and Figure 8: Is it possible to derive the neutral wind from the non-specular meteor echoes by using the method proposed by Oppenheim et al. (2009) and thus demonstrate the neutral wind shear? By using the meteor head echo, the meteoroid properties (e.g., mass, velocity) could also be derived. This provides a good chance to verify the simulation results

---> Response: This is an excellent observation. That would be the next step. It isn't a trivial problem since neutral winds estimates proposed in Oppenheim et al. (2009) need to be validated with other techniques first. We indicated that that is our intention in lines 211-212 of the manuscript.

This reviewer is not satisfied with the authors' response to the above comment. It is not difficult to verify the simulation results based on the cases shown in Figure 8 through a further analysis. Experimental evidence is needed for the simulation results of effects of meteoroid mass and neutral winds on the creation of non-specular meteor trail echoes. This reviewer cannot recommend the paper for publication in its present form.

**Response:**

The comments of this reviewer seem to ignore the main message of the manuscript and bring a criticism to our manuscript as if the subject matter of the paper were the full verification of computer simulations against a specific example given in Figure 8. The full analysis of Figure 8 is in itself another full research paper. These are the reasons why we think this is the case:

1) The reviewer criticizes us for not using the proposed method in Oppenheim et al. (2009) to validate the results shown in Figure 8. The reviewer claims that it is not difficult to verify the simulation results. That criticism is very unfair, very puzzling and out of place since Figure 8 was not the main focus of this manuscript. Clearly, the method described in Oppenheim et al. (2009) to estimate neutral wind values from non-specular reflections is not applicable for the event we observed in Figure 8 since the Oppenheim et al. (2009)'s approach was developed for very short uncoded pulses of $1\mu s$ and an inter-pulse period spanning 60 km (See Oppenheim et al. (2009), Section 2). However, the radar observations shown in Figure 8 were collected using 13-baud Barker code of $1\mu s$ baud length (Section 2, Chau et al. (2007), Section 2). This very clear distinction seems to have been missed by the reviewer.

2) In radar remote sensing research, coded pulses are used to improve the range resolution without losing the maximum average power of the transmitter. In most radar observations, when the targets are assumed quasi-stationary, the decoding procedure is performed by correlating the radar echoes with an identical replica of the code used for transmission (e.g. Farley, 1985a). In the case of meteor-head echo observations, the large Doppler shift does not allow the use of a simple correlation with the transmitter pulse shape (e.g., Wannberg et al., 1996).

3) In Oppenheim et al. (2009), Section 2, first paragraph, third sentence, the authors acknowledge that "*These experiments were unusual in that we used only a short uncoded pulse of 1 microsecond and an inter-pulse period spanning 60 km.*"

4) In addition, the method in Oppenheim et al. (2009) has limited applications and would not work for meteor trails that last less than 3 seconds such as the event shown in Figure 8 of our manuscript. This limitation is stated in Section 3 of Oppenheim et al. (2009), second paragraph: "*While Figure 3 shows a clear overall slope, enabling us to make an accurate determination of the horizontal wind speed, the signal includes substantial deviations from this trend (eg., see the phases changes at 6s). These deviations make it challenging to obtain wind data from meteor trails lasting less than 3 seconds. A close inspection of these deviations shows that they result from coherent phase changes which include as many as a dozen radar pulses. If this resulted from actual motions of the meteor, it would imply that the meteor moves over a km in about 25 ms, a velocity exceeding 50 km/s, clearly an unphysical result.*"

5) Another important limitation in Oppenheim et al. (2009), is indicated in Section 2.2, second paragraph, third sentence: "*Between 94 and 103 km all the*

*meteors give similar velocities at a given height. Above this height, the '07 data shows a considerably larger range of wind speeds. The '07 zonal wind data between 103.5 and 105 km altitude derives from only one meteor and ranges between -80 m/s and +40 m/s, and shows an implausible amount of variability. Above that altitude, between 105 and 109 km, we have 2 meteors which have roughly the same mean of 35 m/s, but one meteor shows quite a lot of shear while the other indicates a relatively constant wind profile. Further experimentation and refinement of the data analysis technique should enable us to improve this…"*

6) In Li et al.(2012), the authors demonstrate experimentally the limitations of the method reported in Oppenheim et al. (2009) to estimate neutral winds. Li et al. (2012) showed that the comparison of the mean meridional neutral wind using Oppenheim et al. (2009)'s method (utilization of non-specular trails) and specular meteor wind generally shows good agreement below 96 km but above 96 km this is not the case. The events we showed in Figure 8 of our manuscript occurred above 96 km. We have added this reference in the manuscript as indicated in item (9), below.

7) Our paper mainly concerns simulations of the effect of neutral winds on the creation of non-specular meteor trail echoes, with research methodologies that are comparable to those reported in: i) Dyrud et al.(2002) L. P., Oppenheim, and ii) Hinrichs et al. (2009), to name a couple.  More important, we believe the research reported in our manuscript will motivate new discussion on the physics of meteors in topics such as fragmentation, sputtering, etc. Due to the nature of our fast simulations, it is possible to construct much needed global studies of the meteor impact on the Earth's Upper Atmosphere.

8) In trying to address the concern raised by this reviewer, we carefully reviewed our radar observations and concluded that it is no possible to use the method reported in Oppenheim et al. (2009) by reasons stated above but also because this method needs to be verified independently with both the Jicamarca High-Power Large Aperture radar and an all-sky meteor radar. In addition, the Oppenheim et al. (2009)'s method needs to be examined carefully when the occurrence of multiple head-echoes and non-specular echoes occur simultaneously, etc. This effort in itself is a big research project that is outside the scope of our manuscript.

9) We have included the following statement near lines 210-212 to capture briefly the reasons stated above: Since the events shown in Figure 8 last less than 3 seconds and therefore the method described in Oppenheim et al., (2009) to estimate neutral winds would not work, we expect to carry out both uncoded and coded radar experiments using Jicamarca High-Power Large Aperture radar and an all-sky meteor radar, to compute neutral wind amplitudes using meteor trails similar to the approach described in (Oppenheim et al., 2009, 2014) and in Li et al. (2012)to establish a complete understanding and characterization of non-specular echoes.

**References**

Chau, J. L. and Galindo, F.: First definitive observations of meteor shower particles using a high-power large-aperture radar, Icarus, 194, 23 – 29, https://doi.org/http://dx.doi.org/10.1016/j.icarus.2007.09.021, http://www.sciencedirect.com/science/article/pii/S001910350700471X, 2008.

Dyrud, L. P., Oppenheim, M. M., Close, S., and Hunt, S.: Interpretation of non-specular radar meteor trails, Geophysical Research Letters, 29, 8–1–8–4, https://doi.org/10.1029/2002GL015953, http://dx.doi.org/10.1029/2002GL015953, 2012, 2002.

Farley, D. T.: On-line data processing techniques for MST radars, Radio Sci., 20, 1177–1184, 1985a.

Hinrichs, J., Dyrud, L. P., and Urbina, J.: Diurnal variation of non-specular meteor trails, Annales Geophysicae, 27, 1961–1967, https://doi.org/10.5194/angeo-27-1961-2009, http://www.ann-geophys.net/27/1961/2009/, 2009.

Li, G., Ning, B., Hu, L., Chu, Y.-H., Reid, I. M., and Dolman, B. K. (2012), A comparison of lower thermospheric winds derived from range spread and specular meteor trail echoes, *J. Geophys. Res.*, 117, A03310, doi:10.1029/2011JA016847.

Oppenheim, M. M., Sugar, G., Slowey, N. O., Bass, E., Chau, J. L., and Close, S.: Remote sensing lower thermosphere wind profiles using non-specular meteor echoes, Geophysical Research Letters, 36, n/a–n/a, https://doi.org/10.1029/2009GL037353, http://dx.doi.org/ 10.1029/2009GL037353, 2009.

Wannberg, G., Pellinen-Wannberg, A., and Westman, A.: An ambiguity-fuction-based method for anlaysis of doppler de- compressed radar signals applied to EISCAT measurements of oblique UHF-VHF meteor echoes, Radio Sci., 31, 497–518, 1996.